# Study of Ni/Y_2_O_3_/Polylactic Acid Composite

**DOI:** 10.3390/ma16145162

**Published:** 2023-07-22

**Authors:** Tilen Švarc, Matej Zadravec, Žiga Jelen, Peter Majerič, Blaž Kamenik, Rebeka Rudolf

**Affiliations:** Faculty of Mechanical Engineering, University of Maribor, Smetanova ulica 17, 2000 Maribor, Slovenia; tilen.svarc1@um.si (T.Š.); matej.zadravec@um.si (M.Z.); z.jelen@um.si (Ž.J.); peter.majeric@um.si (P.M.); blaz.kamenik@um.si (B.K.)

**Keywords:** ultrasound spray pyrolysis, Ni/Y_2_O_3_, lyophilization, PLA, extrusion, injection moulding

## Abstract

This study demonstrates the successful synthesis of Ni/Y_2_O_3_ nanocomposite particles through the application of ultrasound-assisted precipitation using the ultrasonic spray pyrolysis technique. They were collected in a water suspension with polyvinylpyrrolidone (PVP) as the stabiliser. The presence of the Y_2_O_3_ core and Ni shell was confirmed with transmission electron microscopy (TEM) and with electron diffraction. The TEM observations revealed the formation of round particles with an average diameter of 466 nm, while the lattice parameter on the Ni particle’s surface was measured to be 0.343 nm. The Ni/Y_2_O_3_ nanocomposite particle suspensions were lyophilized, to obtain a dried material that was suitable for embedding into a polylactic acid (PLA) matrix. The resulting PLA/Ni/Y_2_O_3_ composite material was extruded, and the injection was moulded successfully. Flexural testing of PLA/Ni/Y_2_O_3_ showed a slight average decrease (8.55%) in flexural strength and a small decrease from 3.7 to 3.3% strain at the break, when compared to the base PLA. These findings demonstrate the potential for utilising Ni/Y_2_O_3_ nanocomposite particles in injection moulding applications and warrant further exploration of their properties and new applications in various fields.

## 1. Introduction

Nickel (Ni) is a strategic metal with catalytic properties, used mainly in organic reactions, since many transformations in organometallic chemistry are catalysed by nickel [1,2,3,4]. Experiments show that the Ni matrix in composite coatings is more active than a pure Ni coating [5,6,7].

Islam et al. [8] and Sun et al. [9] found that oxygen mobility at the surface of nanocrystalline Y_2_O_3_ supports a Ni electrode, which is the most commonly used electrode, plays a crucial role in the oxidative steam reforming of lignocellulosic biomass, or ethanol to hydrogen over a nickel yttrium oxide (Ni/Y_2_O_3_) catalyst. Li et al. discovered that Ni on a Y_2_O_3_ support provides remarkably efficient catalysis for CO_2_ methanation [10]. Similarly, Taherian et al. [11] showed that nickel catalysts on yttria supports are as effective as more expensive commercial catalysts in reforming CO_2_ and methane to syngas.

According to the research work of Guo et al. [12], they showed that catalytic activity is closely related to particle size, which means that smaller particles do not necessarily provide better catalytic activity. It can be concluded that the particle size and the volume-specific surface area are important properties that have a crucial influence on the functional properties of the materials, especially in the case of nanoparticles.

There are many known methods for the synthesis of nanoparticles, but they are generally divided into two groups. The so-called “bottom-up” and “top-down” approaches. One subgroup of the “bottom-up” approach is known as the spray pyrolysis (SP) method. SP methods are widespread in the synthesis processes of various powders and particle suspensions on the microscale and nanoscale. Spray pyrolysis methods consist of five stages: (I) precursor preparation, (II) precursor aerosol generation, (III) aerosol transport, (IV) particle synthesis, and (V) particle collection [13,14]. Precursor solutions that are appropriate for SP methods are metal salts (acetates, bromides, chlorides, hydroxides, nitrates, sulphides) dissolved in water or alcohol [15,16,17]. The precursor characteristics (concentration of the salt, viscosity, density, and surface tension) affect the aerosol size distribution and quantity, which, in turn, has a direct impact on particle size and morphology [18].

Ultrasonic spray pyrolysis (USP) is a subset of spray pyrolysis methods that uses a piezoelectric crystal as the nebuliser for the purpose of aerosol generation. This method was previously used in our research to synthesise different metallic and oxide nanoparticles [15,19,20,21,22,23,24]. Ultrasonic nebulisers have the advantage of narrow aerosol size distribution, which results in well-controlled particle size distributions [14]. Nebulisers based on ultrasound are favoured because of their good energy efficiency in aerosol generation compared to other available techniques [25]. As a result of cavitation and surface waves, standing waves are formed on the fluid’s surface. When the amplitude of the wave is high enough, droplets break off the wave’s peak, resulting in aerosol generation [25]. The precursor aerosols are transported with the help of an inert or reaction carrier gas into the tube reactor, where synthesis occurs. During the particle synthesis, each precursor aerosol is subjected to several physical and chemical processes, such as the evaporation of the solvent liquid, precipitation of the salt, pyrolysis, reduction reactions, and finally, drying of the formatted nanoparticles [13,24]. Due to the high homogeneity of the precursor solution, the created particles have mostly a controlled stoichiometric ratio and morphology [18,26]. High temperatures in the tube reactor cause rapid evaporation of the solvent, which results in high surface-to-volume ratio particles. Collection of the nanoparticles is commonly carried out with gas filtration methods: electrostatic filters, or liquid washing in collection bottles with stabilising agents [24].

The implementation of ultrasound for droplet generation in spray pyrolysis presents an upscale ready process for nanomaterial synthesis, since it operates continuously, and has a good control of the particle size and composition [27,28]. The USP method has a good potential to eliminate technological problems of nanoparticles’ size variations and provides a more controlled nanoparticle synthesis [29,30,31]. An ultrasonic generator is used in this process, which enables the atomization of a solution containing ions of that substance, which are subsequently synthesized into nanoparticles. The atomization of the solution results in the formation of droplets, which are transported to the reaction zone of the USP device, where solvent evaporation, solute reduction, and the formation of nanoparticles take place. In most cases, nanoparticles synthesised by the USP method are collected in the form of a suspension, so it is necessary to dry the suspension to obtain nanoparticles in powder form. The process of lyophilisation is used widely for drying nanoparticles in pharmaceuticals [32,33]. To ensure a successful drying process, the nanosuspensions are dried in multiple steps. This involves a freezing phase, during which the nanosuspension is frozen and the solvent is converted into a crystalline or amorphous solid. Subsequently, the drying phase occurs with a rapid pressure drop in the system. USP coupled with lyophilisation offers a green chemistry approach, as there are no significant pollutants or hazardous chemicals present at the end of the process.

The mechanism of Ni/Y_2_O_3_ nanocomposite particles’ synthesis with USP was proposed in our previous research [15]. In the reactor part of the USP device water evaporation takes place first, and then the dried droplets enter the high-temperature area. Initially, thermal decomposition of the yttrium nitrate and nickel nitrate occurs, leading to the formation of yttrium oxide and nickel oxide. As yttrium oxide is significantly more stable than nickel oxide, a hydrogen reaction can only take place for the formation of nickel. Thus, Ni/Y_2_O_3_ can only be produced following the dehydration and thermal decomposition of metal nitrates, with the hydrogen reduction of nickel oxide being achievable solely in an H_2_/N_2_ atmosphere.

Utilising Ni/Y_2_O_3_ nanocomposites in a polylactic acid (PLA) matrix, with the ability for injection moulding, presents new possibilities for using this composite material for catalytically induced reactions, such as carbon monoxide methanation [8,9]. The injection moulding aspect shows new approaches for producing filters, mesh-like converters, or other complex shapes, where the passing of CO and CO_2_ gases, with the addition of H_2,_ are converted into methane, is used as a measure for removing carbon oxides from process gases. The potential high efficiency of the small Ni/Y_2_O_3_ particle methanation in an injection-moulded PLA/Ni/Y_2_O_3_ composite could be used as an alternative for CO removal from hydrogen-rich gas streams used as fuel for polymer electrolyte fuel cells. Usually, the CO removal is carried out by diffusing the hydrogen-rich gas through a Pd–Ag membrane at high temperatures, or by metal catalysts, such as Au, Pt, Ni, Ru, and Rh, on metal oxide substrates of Al_2_O_3_, SiO_2_, TiO_2_, or ZrO_2_ [34].

The synthesised Ni/Y_2_O_3_ nanocomposite can be used in many fields. One of the applications is the production of ink suitable for application to various surfaces. Such deposits can be used as catalysts in green chemistry applications, since nickel has good catalytic properties in organic reactions [2]. Nickel with the addition of yttrium oxide has been shown to be a good catalyst for the production of hydrogen from ethanol and the methanation of carbon dioxide [35]. Compared to other commonly used catalysts like platinum, nickel delivers a similar performance at a significantly lower cost [36]. In our research, the Ni/Y_2_O_3_ nanocomposite gains potential according to its specific nickel and yttrium properties, to increase the functional properties of PLA matrix as one of the currently most common materials produced from renewable resources. The idea was to use the PLA matrix as a support for catalytically active nanocomposite, so that it would be possible to produce catalytically active layers with various 3D technologies. It was hypothesised that later, an attempt would be made to remove the PLA matrix in such a way that a porous structure of Ni/Y_2_O_3_ nanocomposite with the catalytic activity would be formed. Printing the Ni/Y_2_O_3_ nanocomposite themselves, which would result in the creation of a porous structure, in larger quantities, is currently not feasible [37,38].

## 2. Materials and Methods

### 2.1. Materials

The chemicals used to prepare the precursor solution for the USP synthesis were nickel (II) nitrate (Sigma-Aldrich, Darmstadt, Germany) and yttrium (III) nitrate (Sigma-Aldrich, St. Louis, MO, USA). The carrier (N_2_) and reduction (H_2_) gasses used were 99.999% pure (Messer, Ruše, Slovenia). During the synthesis process, polyvinylpyrrolidone (PVP) (Sigma-Aldrich, Shanghai, China) with an average molar mass of 40,000 g/mol was used to stabilise the nanocomposite particles, and commercially available PLA Ingeo™ Biopolymer 3251D (NatureWorks LLC, Minneapolis, MN, USA) was used as the matrix.

### 2.2. Nanocomposite Particle Synthesis

#### 2.2.1. Ultrasound Spray Pyrolysis

The concentration of nickel(II) nitrate in the precursor solution was 0.025 mol/L, and the concentration of yttrium(III) nitrate was 0.100 mol/L. The solvent was deionised water. The impact of the precursor concentrations was studied previously [15].

The precursor aerosol formation occurred in the ultrasound generator, which uses a piezoelectric crystal to form ultrasonic waves at 1.65 MHz. Nitrogen was used as the carrier gas (1 L/min) and hydrogen as the reaction gas (1 L/min). The reactor tube’s inner diameter was 40 mm, and the length of the evaporation zone and the reaction zone was 1 m. The temperature in the evaporation zone was 200 °C and 900 °C in the reaction zone.

The final size of the nanocomposite particle is strongly dependent on the initial droplet size and precursor solution concentration. The correlation between ultrasonic frequency and the mean droplet diameter was presented by Lang [39], Equation (1). An equation for particle size prediction was developed previously for USP synthesis [40], Equation (2).
(1)$$d=0.34\sqrt[{3}]{\frac{8\pi \gamma }{\rho_{\mathrm{sol}}f^{2}}},$$
(2)$$D=d\sqrt[{3}]{\frac{C_{\mathrm{sol}}M_{\mathrm{par}}}{\rho_{\mathrm{par}}M_{\mathrm{pre}}}},$$

Given that the precursor solution is a low-concentration salt solution, we selected a density and surface tension for the solution equivalent to that of water. To determine the concentration of the precursor solution, hypothetical particle density, molar mass of the precursor, and hypothetical molar mass of the particles, we calculated their mass-averaged values and molar-averaged values. The average particle size was calculated using both mass-averaged and molar-averaged values, which yielded values of 491 nm and 483 nm, respectively. As there was no significant deviation between the two, their average value, 487 nm, will be used for further comparison.

Three collection bottles with 500 mL de-ionised water and PVP were used to collect the synthesised nanocomposite particles. The concentration of PVP was 5.00 g/L. Two batches of USP synthesis were carried out, with each batch taking 3 h to complete.

#### 2.2.2. Lyophilisation

Lyophilisation was performed using an LIO-2000 FLT lyophiliser, produced by Kambič, Slovenia. The lyophilisation protocol used in this study consisted of a freezing period of 5 h and 30 min at −35 °C, followed by a drying period when the pressure was reduced to 0.175 mbar and the temperature was raised to 20 °C. The vacuum at 20 °C was maintained for 20 h to complete the drying process. This protocol was designed carefully to ensure the effective removal of water and preservation of the sample’s structure and properties.

PVP was used as a cryoprotectant and stabiliser during the freeze-drying process, to help preserve the integrity of the Ni/Y_2_O_3_ and prevent agglomeration and sedimentation. It forms a protective layer around the material, shielding it from damage caused by freezing, dehydration, and drying. Additionally, PVP can help maintain the physical and chemical properties of more delicate materials, such as their shape, size, and activity, throughout the drying process [41,42,43].

To ensure consistent drying conditions across all samples, the samples with varying PVP concentrations were freeze-dried in R6 vials equipped with thermocouple attachments, to enable temperature measurement at the bottom of the vial. These measures were taken to minimise variations and ensure accurate and reliable results. In addition, a larger batch of material was dried in a metal tray, to increase the amount of material that could be processed at once.

### 2.3. Composite Preparation

#### 2.3.1. Extrusion Process

The lyophilised Ni/Y_2_O_3_ nanocomposite with 5.00 g/L PVP was hand mixed using a glass rod directly with the PLA granulate, which was dried at 80 °C for 4 h, see Figure 1. The mass ratio of PLA and Ni/Y_2_O_3_ was 100:2. The extrusion process was carried out in two steps using a parallel twin-screw extruder (Thermo Fisher Scientific, Waltham, MA, USA) with an L/D of 40. Extrusion and compounding were performed using the same parameters. The temperature at the feed zone was set to 60 °C and the subsequent 7 zones increase in temperature with an increment of 5 °C, from 180 °C to 210 °C, with the die heated to 180 °C. The material was processed with 100 rpm screw speed and extruded through a 2 mm circular die as a filament and air-cooled on a 40 cm long conveyor belt.

In the first step of compounding the PLA granulate and Ni/Y_2_O_3_ nanocomposite took place, followed by the extrusion of the previously prepared PLA/Ni/Y_2_O_3_ composite granules into a filament. The two-step extrusion ensured a homogenised material. The same extrusion process was repeated for the pure PLA granulate, with the goal to obtain a reference material.

#### 2.3.2. Injection Moulding

Thermo Scientific HAAKE MiniJet (Walthan, MA, USA) Pro was used to produce the bending specimens. Pure PLA granules and PLA/Ni/Y_2_O_3_ granules were used to prepare the three-point flexural test specimens in accordance with ISO 178 [44]. The following parameters were used in the injection moulding for both types of samples: T_melting_ = 190 °C and T_mould_ = 50 °C; while the injection pressure was 700 bar and the holding pressure was 450 bar. The injection time and the holding time were both 5 s.

### 2.4. Nanocomposite Characterisation Methods

#### 2.4.1. Transmission Electron Microscopy

Transmission Electron Microscopy (TEM) with Energy Dispersive X-ray (EDS) (Oxford Instruments, Abingdon, UK) was used to evaluate the state of the nanocomposite particles in the lyophilised state. A Jeol JEM 2100 (Jeol, Tokyo, Japan) was used with a 200 kV LaB_6_ electron source.

The size distribution was obtained by measuring the particle size on the TEM images. A total of 200 particles were measured. The data were evaluated with ImageJ software version 1.53t [45].

#### 2.4.2. Viscosity and Drying Times

The viscosity of the nanocomposite suspensions with different PVP concentrations was measured using MCR with a CP50-0.5 D measuring cone (Anton Paar, Graz, Austria). In total, 1100 mL of each sample that was maintained at 25 °C was used for the measurements. The viscosity was evaluated in the shear rate interval of 1–100 1/s and Newton’s regression was used to determine the final viscosity. For each sample, 3 measurements were conducted. The Ni/Y_2_O_3_ nanocomposite suspension, obtained via USP synthesis, had PVP added to achieve the following PVP concentrations: 2.50 g/L, 5.00 g/L, 10.00 g/L, and 20.00 g/L. In addition, a water-based solution containing PVP at the same concentrations was used as a baseline for comparison with the nanocomposite suspension, to assess the influence of the nanocomposite particles on the suspension. The same Ni/Y_2_O_3_ nanocomposite suspension was used to determine the length of the drying process.

### 2.5. PLA Ni/Y_2_O_3_ Composite Characterisation Methods

#### 2.5.1. Mechanical Properties

A three-point flexural test was performed to compare the mechanical properties of the PLA/Ni/Y_2_O_3_ composite with the base PLA. The test was conducted on a ZWICK/ROELL Z010 (Zwick Roell Group, Ulm, Germany) materials testing machine in accordance with ISO 178, where flexural strength, flexural stress, and flexure were measured. Six parallels were tested for each type of material. The length of the samples was 80 mm, width 4 mm, and height 10 mm. The preload was 0.1 MPa and the test speed was 10 mm/min.

#### 2.5.2. Scanning Electron Microscopy

A scanning electron microscope (SEM), Sirion 400 NC (FEI Sirion 400 NC, FEI Technologies Inc., Hillsboro, OR, USA), with an EDS INCA 350 (Oxford Instruments, UK), was used for the investigations of the fracture surface for the prepared PLA/Ni/Y_2_O_3_ composite and pure PLA. The samples for SEM examination were prepared by breaking directly in liquid N_2_, thus preserving the authenticity of the surface for microscopic observation. The investigated surfaces of the samples were sputter coated with Au for 60 s, in order to produce a conductive film and improve the SEM imaging of the samples.

## 3. Results and Discussion

### 3.1. Transmission Electron Microscopy

The TEM observations revealed a high degree of roundness for the Ni/Y_2_O_3_ nanocomposite particles nanoparticles at lower magnifications as shown in Figure 2. The smallest observed non-agglomerated nanoparticle had a diameter of 55 nm, while the largest had a diameter of 1603 nm. The average particle size estimated by the number of nanoparticles, was measured to be about 466 nm. The relative frequency of particles by size is presented in Figure 3. Slightly over half of the particles are in the size range of 100 nm to 500 nm, while 34% fall between 500 nm and 1000 nm. The smallest particles, below 100 nm, comprise only 5.2% of the total. The remaining particles are larger than 1000 nm.

The calculated particle diameter obtained using Equations (1) and (2), which was 487 nm, and the measured average particle diameter of 466 nm, show good agreement. This confirms that the equations presented previously in [39] can be used to predict the particle size of Ni/Y_2_O_3_ nanocomposite particles. Further study is needed to determine the applicability to the general area of metallic and ceramic composite particles produced by USP.

At higher magnifications in TEM investigations, a wavy surface of the nanoparticles could be observed, as can be seen in Figure 4. This is probably the result of the sintering of smaller nanoparticles in the size of 10 nm, which form round Ni/Y_2_O_3_ nanocomposite particles. Moreover, Figure 4 provides a detailed view of the crystal lattice on the surface of the Ni/Y_2_O_3_ nanocomposite particles. By comparing the measured distance between the crystal planes, which was 0.344 nm and 0.343 nm, with the theoretical distance between crystal planes in cubic face-centred nickel, which was 0.348 nm, we can confirm the presence of cubic nickel on the particle surface. This information confirms the mechanism established previously [15], which proposed the formation of elemental nickel on the surface of the Ni/Y_2_O_3_ nanocomposite particles.

Figure 5 shows the electron diffraction of the Ni/Y_2_O_3_ nanocomposite particles. A great agreement can be observed between the experimental and the theoretical diffraction image of yttrium oxide [46]. Therefore, the presence of yttrium oxide in the core of the nanocomposite can be confirmed.

EDS analysis was performed to investigate the elemental composition of individual Ni/Y_2_O_3_ nanocomposite particles. Table 1 presents the results of the performed analysis based on Figure 2. No significant difference in the chemical composition of smaller and larger particles can be observed. By comparing the ratio between nickel nitrate and yttrium nitrate in the precursor solution and the ratio between nickel and yttrium in the particles obtained from the EDS analysis, we can draw the conclusion that there is some material loss observed specifically on the nickel side during the USP synthesis. This loss may potentially occur due to deposition on the reactor walls and variations in the reaction rates between nickel nitrate and yttrium nitrate.

### 3.2. Viscosity and Drying Time

Viscosity has a direct effect on the rate of nanoparticle settling. Higher viscosity results in slower nanoparticle settling rates, and lower viscosity results in faster settling rates. The viscosity of the fluid medium has a major influence on the drag force experienced by nanoparticles, which affects the settling velocity of the particles directly. As viscosity increases, the drag force experienced by the nanoparticles also increases, causing them to settle at a slower rate [47]. When PVP is added to water, it increases the viscosity of the solution. This is due to the formation of a three-dimensional network of polymeric chains, which leads to increased interactions between water molecules and reduced mobility of the molecules. The viscosity is increased as a result [48,49].

The viscosity values of the suspensions at different PVP concentrations are presented in Table 2. The addition of Ni/Y_2_O_3_ nanoparticles did not impact the viscosity of the suspensions significantly, while the viscosity was affected significantly by the concentration of PVP.

Higher viscosity solutions tend to require longer drying times, resulting in reduced product yields. Therefore, it is important to control the viscosity of the solution, in order to optimise the lyophilisation process. The figures show the effect of PVP concentration in water on the viscosity of it, as well as the impact of the presence of Ni/Y_2_O_3_ nanoparticles on the viscosity.

The freezing time was not affected significantly by varying the concentrations of PVP. However, the drying time was impacted notably, as illustrated in Figure 6. The drying time was defined as the duration between the temperature increase and the point when the sublimation front reached the bottom of the vial and each suspension are shown in Table 3. The impact of additional PVP in the solution was more pronounced at lower PVP concentrations, 2.50 g/L and 5.00 g/L, while additional increases do not affect the drying times significantly.

The results show that the drying time increased as the concentration of PVP increased from 2.50 g/L to 5.00 g/L, indicating that lower PVP concentrations have a shorter drying time. However, as the PVP concentration increased to 10.00 g/L and 20.00 g/L, the drying time did not exhibit significant changes, indicating that there might be an optimal concentration range for PVP that balances the cryostabilisation effect, stabilisation effect, and drying time. From our range of data, it was indicated that this concentration is close to 5.00 g/L; therefore, this concentration of PVP was used in the Ni/Y_2_O_3_ suspensions in all subsequent freeze-drying processes and the PLA/Ni/Y_2_O_3_ composite preparation.

### 3.3. Mechanical Properties

Flexural testing of Ni/Y_2_O_3_ incorporated into PLA showed a slight average decrease (8.55%) in flexural strength and a small decrease, from 3.7 to 3.3%, in strain at the break, when compared to the base PLA (results in Table 4). The addition of Ni/Y_2_O_3_ into a PLA matrix has a minimal effect on its mechanical properties, with the strain–stress curve of PLA/Ni/Y_2_O_3_ (Figure 7), trailing the PLA curve closely. However, these slightly lower values may indicate Ni/Y_2_O_3_ accelerated degradation. The lower strain at break is typical of the degradation-induced embrittlement of polymers [50,51,52]. As degradation processes occur, polymer chains may break, cross-linking between chains may weaken, or molecular weight may decrease. These changes in the molecular structure typically result in a decrease in the polymer’s ability to elongate or deform plastically, leading to embrittlement [53,54]. While these processes typically already take place during the processing of polymers, it usually takes a number of consecutive processing steps, for these changes to become apparent [55,56]. Studies have found that PLA usually experiences low degradation within 1 to 3 reprocessing cycles [57].

### 3.4. Scanning Electron Microscopy

SEM examination of the fracture surfaces revealed that the Ni/Y_2_O_3_ particles were distributed uniformly throughout the volume of the PLA matrix. Figure 8 shows the fracture of both samples at three different magnifications, identifying individual agglomerated groups of Ni/Y_2_O_3_ particles in the PLA matrix. The comparison of the fracture surfaces of PLA and PLA/Ni/Y_2_O_3_ did not show similar characteristics, as the fracture facets in the case of pure PLA were significantly longer, which means that the fracture in the PLA test tube was tougher compared to the PLA/Ni/Y_2_O_3_ composite. Namely, in the case of PLA/Ni/Y_2_O_3,_ the SEM examination revealed significantly shorter fracture facets and smaller fracture surfaces, indicating that the fracture was more brittle, which agrees with the results of the flexural test.

These findings demonstrate the potential for utilising Ni/Y_2_O_3_ nanocomposite particles, prepared with a green chemistry technique, in injection moulding applications, and warrants further exploration of their properties and potential applications in various fields.

The successful synthesis and characterisation of Ni/Y_2_O_3_ nanocomposite particles using the USP method provides a new avenue to produce high-quality materials with potential applications in various industries. Additionally, the Ni/Y_2_O_3_ nanocomposite particle suspensions were lyophilized, to obtain a dried material that is suitable for incorporating into a suitable polymer matrix, such as PLA, that can be extruded into a 3D-print-ready filament.

In further research, we will focus on determining the catalytic properties of Ni/Y_2_O_3_ nanocomposites and PLA/Ni/Y_2_O_3_ composites, as previous research with X-ray photoelectron spectroscopy has shown non-stoichiometry [15].

## 4. Conclusions

Overall, this study contributes to the field of nanocomposite particle synthesis, PLA-based filament extrusion, and injection moulding technique. The following conclusions can be drawn from our research:The USP method proved to be a highly effective green chemistry approach in the successful synthesis of Ni/Y_2_O_3_ nanocomposite particles.The use of modified, previously presented equations, allowed for accurate size prediction of the nanoparticle synthesised by USP, which was confirmed by TEM analysis.The presence of the Y_2_O_3_ core and Ni shell was confirmed with TEM and electron diffraction.The proper concentration of PVP (5 g/L) in the Ni/Y_2_O_3_ nanoparticle suspension before lyophilisation leads to optimal cryostabilisation effects, stabilisation effects and drying times.The PLA/Ni/Y_2_O_3_ composite material was extruded successfully, so that it was possible to prepare flexural samples by injection moulding.The addition of Ni/Y_2_O_3_ into a PLA matrix has minimal effect on its flexural properties, with the strain-stress curve of PLA/Ni/Y_2_O_3_ being similar to that of pure PLA.The Ni/Y_2_O_3_ particles were practically uniformly distributed throughout the entire volume of the PLA matrix.

## Figures and Tables

**Figure 1 materials-16-05162-f001:**
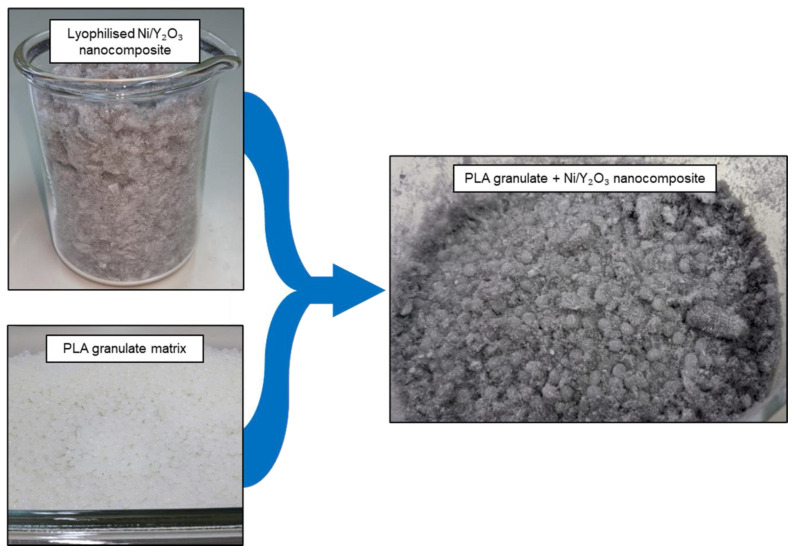
PLA granulate with Ni/Y_2_O_3_ nanocomposite particles.

**Figure 2 materials-16-05162-f002:**
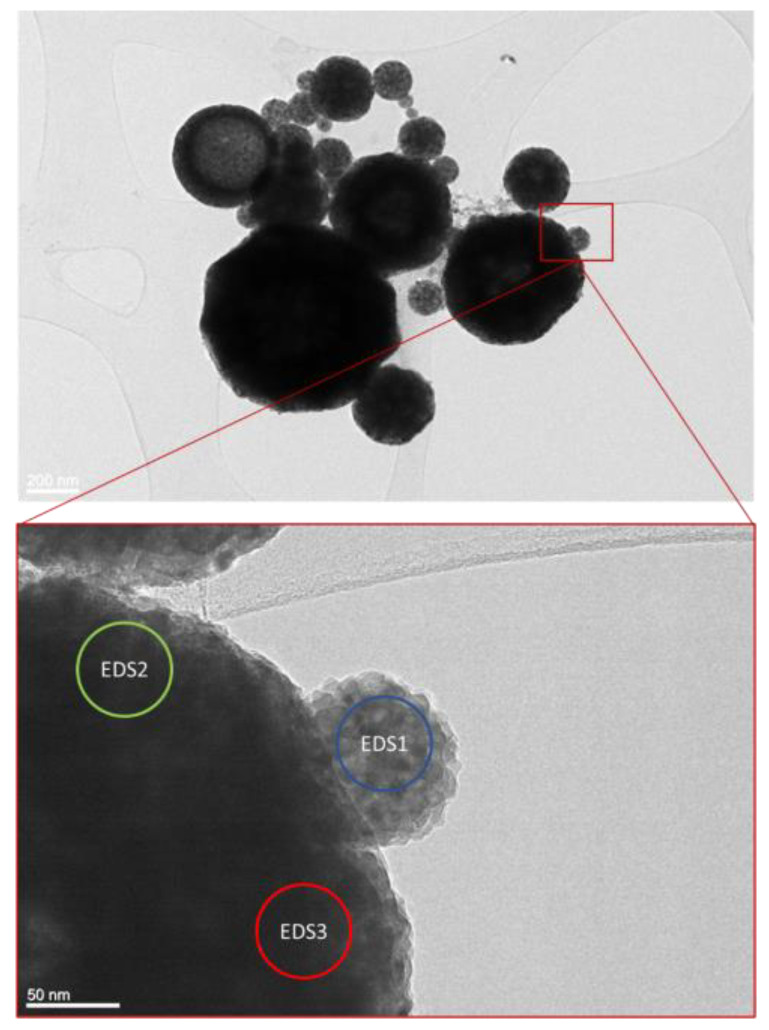
TEM microstructure of the Ni/Y_2_O_3_ nanocomposite particles.

**Figure 3 materials-16-05162-f003:**
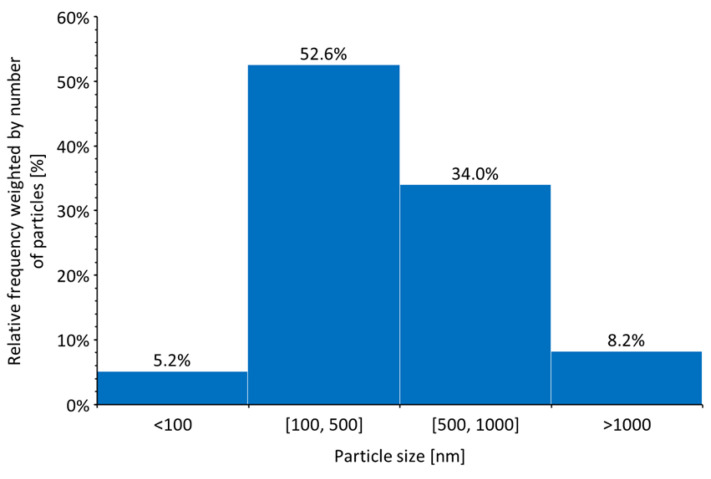
Ni/Y_2_O_3_ nanocomposite particle size distribution weighted by number of particles; obtained by measuring the diameter of 200 particles.

**Figure 4 materials-16-05162-f004:**
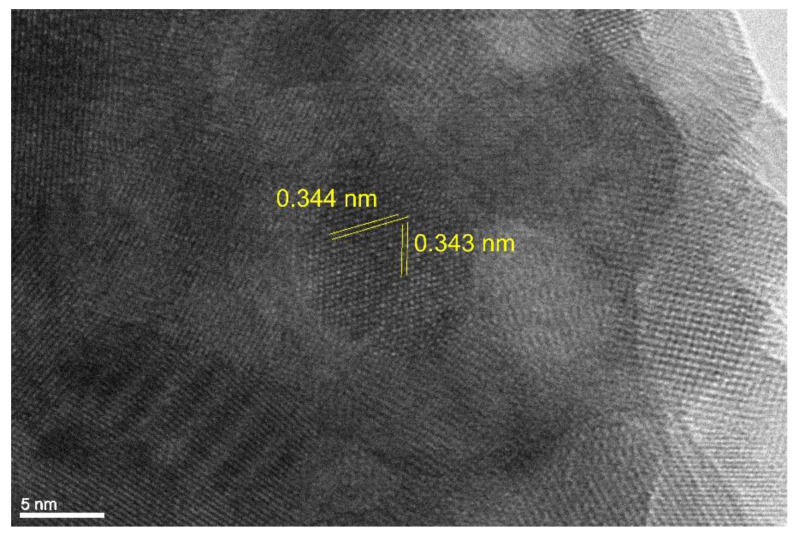
Crystal lattice structure on the Ni/Y_2_O_3_ nanocomposite particle surface.

**Figure 5 materials-16-05162-f005:**
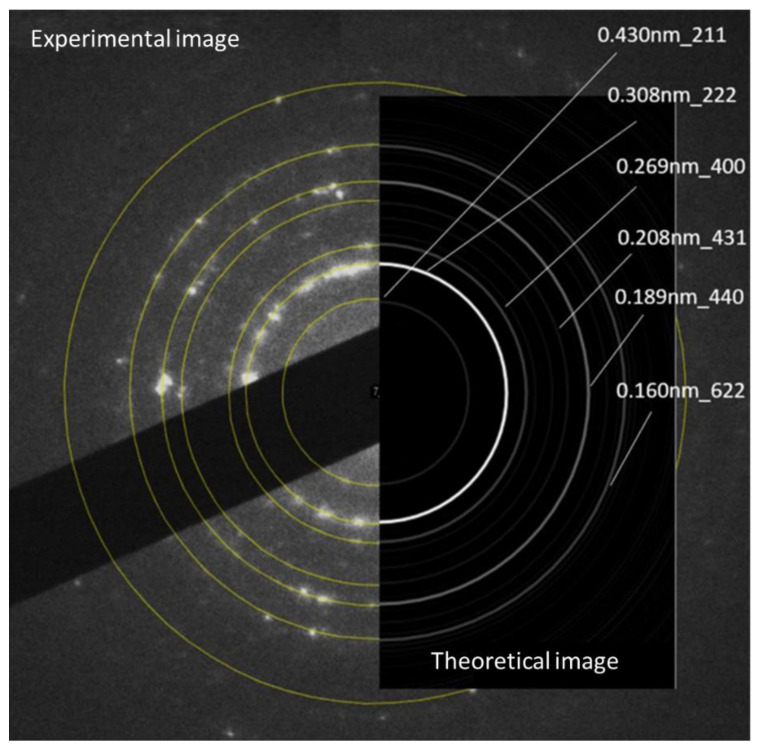
Electron diffraction of the Ni/Y_2_O_3_ nanocomposite particles; experimental and theoretical image.

**Figure 6 materials-16-05162-f006:**
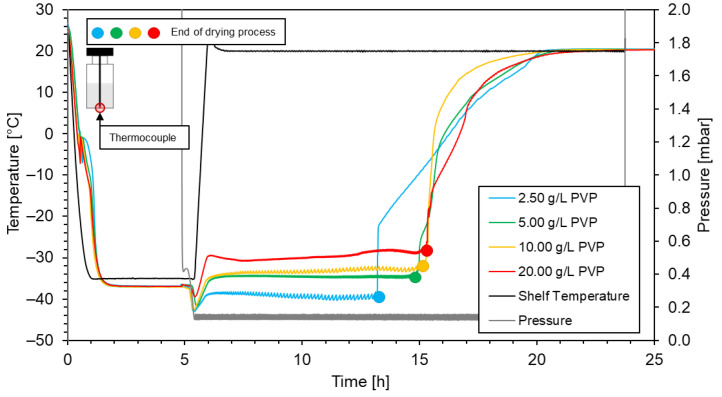
Impact of PVP concentration on the drying time of the Ni/Y_2_O_3_ nanocomposite particle suspension.

**Figure 7 materials-16-05162-f007:**
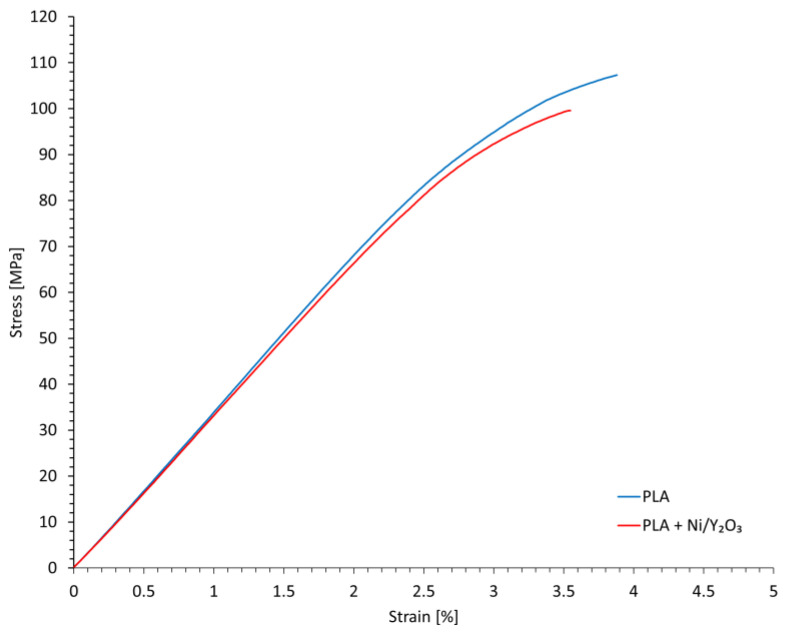
Representative flexural strain–stress curves of PLA and PLA/Ni/Y_2_O_3_ composite.

**Figure 8 materials-16-05162-f008:**
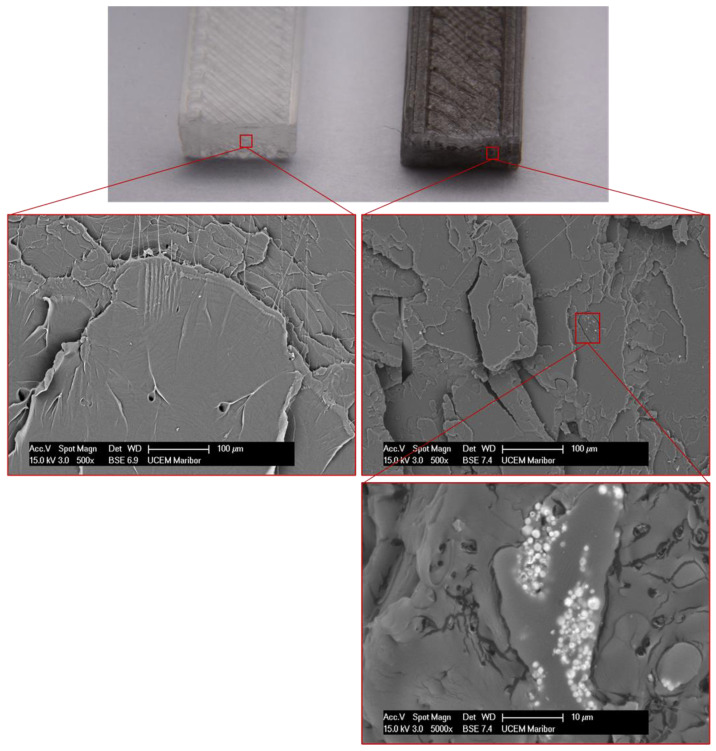
SEM microstructure of PLA—Ni/Y_2_O_3_ fracture and pure PLA at 500× and 5000× magnification.

**Table 1 materials-16-05162-t001:** EDS analysis of the Ni/Y_2_O_3_ nanocomposite particles.

Element	EDS 1 [at. %]	EDS 2 [at. %]	EDS 3 [at. %]
O	54.09 ± 0.01	51.35 ± 0.01	57.95 ± 0.01
Ni	5.16 ± 0.04	7.19 ± 0.04	3.04 ± 0.04
Y	40.75 ± 0.01	41.46	39.01 ± 0.01
Total	100.00	100.00	100.00

**Table 2 materials-16-05162-t002:** Impact of PVP and Ni/Y_2_O_3_ nanocomposite particle on the suspension viscosity.

PVP Concentration [g/L]	Viscosity of H_2_O + PVP [mPa·s]	Viscosity of Ni/Y_2_O_3_ + H_2_O + PVP [mPa·s]
0.0	0.80 ± 0.05	0.82 ± 0.04
2.5	0.90 ± 0.01	0.92 ± 0.04
5.0	0.98 ± 0.05	0.98 ± 0.04
10.0	1.10 ± 0.07	1.05 ± 0.06
20.0	1.36 ± 0.05	1.42 ± 0.13

**Table 3 materials-16-05162-t003:** Impact of PVP concentration on the drying time of the Ni/Y_2_O_3_ nanocomposite particle suspension.

PVP Concentration [g/L]	Drying Time
2.50	7 h 36 min
5.00	9 h 06 min
10.00	9 h 30 min
20.00	9 h 48 min

**Table 4 materials-16-05162-t004:** Average flexural test results.

	PLA	PLA/Ni/Y_2_O_3_
Flexural strength [MPa]	106.37 ± 1.31	97.28 ± 2.74
Flexural stress at break [MPa]	106.37 ± 1.31	97.28 ± 2.74
Strain at break [%]	3.7 ± 0.18	3.3 ± 0.23

## Data Availability

Data available on request from the corresponding author.

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
