# Peer review of "Study of Ni/Y2O3/Polylactic Acid Composite"

_materials, 2023, doi:10.3390/ma16145162_

Round 1
Reviewer 1 Report (New Reviewer)
Rudolf and coworkers report the synthesis of Ni/Y2O3 nanoparticles by using ultrasonic spray pyrolysis. With the stabilization of PVP, the polymer-NPs hybrids incorporate into PLA to examine the effects on the mechanical properties. The manuscript is well-established, but some important points should be addressed:
1. In Figure 3, how many particles the authors used for counting the size distribution?
2. The particle size is not uniform, and what are the benefits of synthesizing the particles by the USP method?
3. In experiment 2.2, did the author do any purification procedures after pyrolysis? At the same time, how did the authors confirm the formation of Y2O3 and Ni? XRD data need to be present to confirm the phase of inorganic materials. The element analysis on SEM only tells what the element is, but it can not tell the formation of the new material.
4. In mechanical testing, how many repeating times of the tensile testing? Please show the error bar and standard deviation because the author shows a very small difference in the mechanical properties. It can not tell the increasing/decreasing properties without presenting an error bar.
Author Response
Review 1
Rudolf and coworkers report the synthesis of Ni/Y2O3 nanoparticles by using ultrasonic spray pyrolysis. With the stabilization of PVP, the polymer-NPs hybrids incorporate into PLA to examine the effects on the mechanical properties. The manuscript is well-established, but some important points should be addressed:
- In Figure 3, how many particles the authors used for counting the size distribution?
In total the diameter of 200 particles was measured, as is mentioned in chapter 2.4.1. The number of particles will be added in the description of Figure 3 for improved clarity.
- The particle size is not uniform, and what are the benefits of synthesizing the particles by the USP method?
USP is a continuous process that gives a sufficiently narrow size distribution. The energy-efficiency of ultrasound nebulisers is favourable compared to other available techniques. Additional USP is a upscale ready process, that enables a immediate steric stabilisation.
- In experiment 2.2, did the author do any purification procedures after pyrolysis? At the same time, how did the authors confirm the formation of Y2O3 and Ni? XRD data need to be present to confirm the phase of inorganic materials. The element analysis on SEM only tells what the element is, but it can not tell the formation of the new material.
No purification was done after the pyrolysis. Y2O3 was confirmed by TEM diffraction on figure 5. Standard XRD diffraction is not possible with the nanomaterial prepared via USP synthesis as the amount of material needed to obtain the data exceeds the amount that can be sensibly prepared with USP.
- In mechanical testing, how many repeating times of the tensile testing? Please show the error bar and standard deviation because the author shows a very small difference in the mechanical properties. It can not tell the increasing/decreasing properties without presenting an error bar.
During the bending testing six repetitions were made, which is stated in chapter 2.5.1. The standard was added in all tables where it is sensible. Adding error bars on the Stress-Strain curve is not sensible since the plots shown are representative and not averaged. The Stress-Strain curve is added into the manuscript to give the reader a general idea how the composite Ni/Y2O3 + PLA material behaves in comparison with pure PLA. For a comprehensive understanding of the material properties table 4 should be noted.

Reviewer 2 Report (New Reviewer)
Authors tried to show how to prepare nano particles and the effect of compounding those with PLA on properties.
General comments: Lots of detailed description on methods is missing including the number of repeats (in Chap 2). All the figures and tables should show statistical analysis with standard deviations (in Chap 3).
Line 8: needs to specify the equation.
Lines 16-19, 348: It seems that the mechanical properties become worse if the particles are added. Then, why is there potential? Authors did not show why compounding the nanoparticles has some potential. This is the most important weakness of the manuscript.
Line 116: More information is required such as purity and molecular weights (Mw and Mn) of the polymers. Gases (N2, H2) should be added here as well.
Line 117: What does USP synthesis mean?
Eqs. 1 and 2: Need to define terms and to show how to obtain those.
Line 173: how to mix?
Line 176: more information for extrusion is necessary such is L/D, temperatures, rpm, die size, cooling method, etc. in each extrusion.
Fig. 1: Higher quality is necessary.
Line 203: More details for rheometry is necessary such as temperature, fixture, shear rate, etc.
Line 212: More details for the tests are necessary such as sample dimensions and speed.
Line 219: Any coating?
Line 257: where is the theoretical diffraction image comping from?
Line 284: There should be supporting paper.
Line 288: What does “water” mean? Any temperature and shear rate information?
Line 296: Need to correct figure number.
Lines 323-324: Need to show more discussion regarding “degradation”
There are many places where unnecessary capitalization was used such as Poly..., Transmission, Particle, Spray, etc. there is also Italicization issue in equations such is Eqs. 1 and 2 and Lines 187 & 188. Authors should revise all those.
Author Response
Review 2
Authors tried to show how to prepare nano particles and the effect of compounding those with PLA on properties.
General comments: Lots of detailed description on methods is missing including the number of repeats (in Chap 2). All the figures and tables should show statistical analysis with standard deviations (in Chap 3).
Line 8: needs to specify the equation.
The sentence which mentions the equation in the abstract was removed to not cause confusion.
Lines 16-19, 348: It seems that the mechanical properties become worse if the particles are added. Then, why is there potential? Authors did not show why compounding the nanoparticles has some potential. This is the most important weakness of the manuscript.
Testing of mechanical properties was performed to understand how the composite will behave during and after printing, although mechanical properties are not considered crucial in the proposed use of Ni/Y2O3 nanoparticles.
The idea was to use the PLA matrix as a support for catalytically active nanoparticles, so that it would be possible to produce catalytically active layers with various 3D technologies. It was hypothesized that later an attempt would be made to remove the PLA matrix in such a way that a porous structure of nanoparticles with the catalytic activity would be formed. Printing the nanoparticles themselves, which would result in the creation of a porous structure, in larger quantities, is currently not feasible. New text was added in the Introduction section (lines 117-123), along with two references (37,38).
Line 116: More information is required such as purity and molecular weights (Mw and Mn) of the polymers. Gases (N2, H2) should be added here as well.
Additional information was added to the chapter as recommended by the reviewer.
Line 117: What does USP synthesis mean?
USP synthesis represents one of the bottom-up methods of nanoparticle synthesis. An ultrasonic generator is used in this process, which enables the atomization of a solution containing ions of that substance, which are subsequently synthesized into nanoparticles. The atomization of the solution results in the formation of droplets, which are transported to the reaction zone of the USP device, where solvent evaporation, solute reduction and the formation of nanoparticles take place. These are collected in different systems, most often in bottles with the selected medium. According to the request, an additional description of the USP method has been inserted into the manuscript in the Introduction section (lines 74-78).
Eqs. 1 and 2: Need to define terms and to show how to obtain those.
The terms are defined in at the end of the manuscript. The terms represent material properties that are widely available.
Line 173: how to mix?
An additional explanation was added.
Line 176: more information for extrusion is necessary such is L/D, temperatures, rpm, die size, cooling method, etc. in each extrusion.
Additional information was added to the chapter as recommended by the reviewer.
Fig. 1: Higher quality is necessary.
The figure was enlarged, and a higher quality image was used.
Line 203: More details for rheometry is necessary such as temperature, fixture, shear rate, etc.
Additional information was added to the chapter as recommended by the reviewer.
Line 212: More details for the tests are necessary such as sample dimensions and speed.
Additional information was added to the chapter as recommended by the reviewer.
Line 219: Any coating?
Additional information was added to the chapter as recommended by the reviewer.
Line 257: where is the theoretical diffraction image comping from?
Additional reference was added.
Line 284: There should be supporting paper.
The information is in regard to table 2. The sentences were rewritten to better illustrate the connection.
Line 288: What does “water” mean? Any temperature and shear rate information?
The table headers were changed to better illustrate the measured samples. Additional information was added in chapter 2.4.2.
Line 296: Need to correct figure number.
Figure number was corrected.
Lines 323-324: Need to show more discussion regarding “degradation”
Additional discussion regarding degradation was added as recommended by the reviewer.
Comments on the Quality of English Language
There are many places where unnecessary capitalization was used such as Poly..., Transmission, Particle, Spray, etc. there is also Italicization issue in equations such is Eqs. 1 and 2 and Lines 187 & 188. Authors should revise all those.
Italicization was removed.

Round 2
Reviewer 1 Report (New Reviewer)
Accept the current format.
Reviewer 2 Report (New Reviewer)
Please Italicize all the symbols in equations.
Please Italicize all the symbols in equations.
This manuscript is a resubmission of an earlier submission. The following is a list of the peer review reports and author responses from that submission.
Round 1
Reviewer 1 Report
Quantitative results are insufficient in the abstract, and most presented results are generalizations repeated in all previous references. First of all, the purpose of conducting research and innovation should be presented transparently and all results should be presented quantitatively. In the abstract, you can use abbreviations, especially that presented in the title or keywords (such as FDM).
What is the reason for using filler and fabrication composite (addition of lyophilized Ni/Y2O3 into the PLA matrix)? What is the justification for fabrication of a composite when the strength is reduced compared to the PLA sample? The reason for the decrease in tensile strength should also be mentioned.
The introduction is very general. Although the introduction is long, it is written superficially in some paragraphs. Also, in the end, a suitable summary of the importance of the present issue should be provided. Also, The number of reviewed articles in the introduction is minimal.
The introduction should be written better and needs minor revisions. Use the following resources to deepen the introduction. A New Strategy for Achieving Shape Memory Effects in 4D Printed Two-Layer Composite Structures. 4D printing of PLA-TPU blends: effect of PLA concentration, loading mode, and programming temperature on the shape memory effect. 4D printing of polyvinyl chloride (PVC): A detailed analysis of microstructure, programming, and shape memory performance. The first few paragraphs of the introduction, which are related to nickel and its coating, are far from the article's main topic, and you can summarize or delete them.
Referencing the articles is awful and disappointing (Lines 33, 34, and 62). The use of general sentences with more than four references can be seen in all parts of the introduction. On the other hand, appropriate references were not used to analyze the results.
It is required that all printing parameters (such as speed, layer thickness, nozzle temperature, bed temperature, and nozzle diameter) should be summarized in a table. One of the most important parts of 3D printing new composites is providing optimal printing parameters, which has been forgotten in this work.
The image quality is low, and higher-quality images with a higher resolution should be used—for example, the images in Figure 1. The average grain size is calculated by what method? The error bar should be added to Figure 3.
The results section is well organized and categorized. But some parts are just reporting the results, which require corrections and deepening the analysis and discussion. Use the suggested resources to deepen the discussion. Statistical and experimental analysis of process parameters of 3D nylon printed parts by fused deposition modeling: response surface modeling and optimization. Development of Pure Poly Vinyl Chloride (PVC) with Excellent 3D Printability and Macro‐and Micro‐Structural Properties. It is suggested to modify the conclusion section as well as the abstract.
The images presented in Figure 9 are used in raw form. Additional explanations are not provided in the text. Add scale bar and label for additional description. Why are the results of the tensile test and DMA not consistent? Has adding Ni/Y2O3 into the PLA decreased the elastic modulus and increased the storage modulus? How has the reproducibility of the results been checked?
No Comment.